# Engaging with the State
# Illegalized migrants, welfare institutions and the law in French-Speaking Belgium

Sophie Andreetta*

FRS-FNRS Research Fellow, University of Liège
* sandreetta@uliege.be

Monday, January 31, 22

## Abstract

**By highlighting informal strategies and solidarities on the one hand, and protests on the other, current studies of citizenship and illegalized migration describe two main forms of political subjectivities among illegalized migrants: 'without' the state or 'against' it. In contrast, this article unpacks how migrants make use of state laws and institutions, voice expectations, and pursue their claims using official venues – in short, how they act 'with' the state. It builds on ethnographic fieldwork on illegalized migrants' welfare requests in French-speaking Belgium and the various sets of actors involved in assessing or furthering their cases. Migrants' discourses and expectations of the welfare office provide insights into their understandings of the state, highlight the crucial role of immigration lawyers in brokering cases, and ultimately allow for a more nuanced reading of the idea that welfare dependency leads to deportability. On a theoretical level, this article contributes to ongoing debates in the study of statehood and in migration studies, showing how procedural fairness can be a central aspect of migrants' relationship to and expectations of the state, and how migrants' strategies leading towards inclusion can be formal ones, based on existing state laws and institutions.**

## 1. Introduction

**January 2019, Brussels**

It is 7:30 in the morning; Salim and I meet at the train station. I am there to show him the way to the welfare office and to help him ask for state-covered medical assistance. He has only been in Belgium for a few weeks, and does not know the city yet.

Salim and I met a couple of days earlier at the NGO where I was volunteering. Salim came to see a social worker and showed her a 'certificate for emergency medical assistance', which he had been given at the hospital the day before. The document stated that he needed surgery and a short-term hospital stay. On the envelope, the doctor wrote the welfare office's address. Salim does not speak French nor does he have a pre-paid Belgian SIM card

that would allow him to use his smartphone to look up the welfare office's location. He asked if someone could go with him. I immediately volunteered, explaining that it would help with my research. (Excerpt from fieldnotes, January 2019).

Current scholarship on irregular migration, which mainly focuses on the vulnerabilities of migrants having a precarious legal status, concentrates on migrants' daily tactics and informal strategies of survival outside of or hidden from the state (Coutin 2005; Psimmenos & Kassimati 2006; Vasta 2011; Le Courant 2016). In contrast, this article explores the ways in which illegal(ized) migrants mobilize state laws and public institutions – such as welfare bureaucracies and courts – to gain access to health care or financial support.

The migrants that I studied were consistently referred to (by lawyers and social workers alike) and self-identified as 'illegal' or 'illegal residents'. Throughout this article, I will refer to them as 'illegalized' to highlight the legal and political processes that allow for them to be identified as 'illegal' (De Genova 2002). Based on my own ethnographic investigations of these migrants' interactions with social assistance bureaucracies in French-speaking Belgium, and drawing on ethnographic studies of statehood (Thelen et al. 2018; Blundo & Le Meur 2009; Bierschenk & Olivier de Sardan 2014) and on sociolegal studies (Merry 1990; Ewick & Silbey 1998), I reflect on illegalized migrants' relationship to and engagement with the state (Thelen et al. 2018).

Existing scholarship indeed analyses 'ordinary' citizens' interactions with civil servants (Thelen et al. 2018; Blundo & Le Meur 2009; Bierschenk & Olivier de Sardan 2014) and the ways in which they 'understand and use the law' (Merry 1990: 5), taking gender, social class, or race into account (Nielsen 2000). This article aims to further these debates by focusing on illegalized migrants' right to social assistance. In doing so, it contributes to recent studies on the daily practices of immigration bureaucrats (Spire 2008; Eule et al. 2017, 2019; Tuckett 2018; Ratzmann 2021), which rarely consider migrants' expectations, their knowledge of their rights, or their understandings of their relationship to the state. It illuminates how, in addition to having recourse to informal or non-state institutions such as NGOs, illegalized migrants use state laws, legal aid clinics and courts to negotiate access to certain public services, forcing welfare bureaucracies to comply with the procedural and substantive principles underlying the provision of social assistance. Migrants' lawyers regularly take the administration to court, asking that the state be forced to respect its own laws, which stipulate that requests be processed in a timely manner, that administrative decisions be justified in both law and by the facts of the case, and that health care and financial aid be provided to those at risk of experiencing inhumane living conditions.

This article therefore contributes to ongoing debates both in the study of statehood and in migration studies by showing how procedural fairness can be a central aspect of – in this case – migrants' relationship to and expectations of the state, and how illegalized migrants' strategies and practices to be included can (sometimes) be formal ones, based on state laws and institutions.

In Belgium, welfare administrations are tasked with delivering social assistance, which is meant to safeguard human dignity as defined in the Constitution. Since 1996, social assistance has been restricted to emergency medical assistance (EMA) for people staying in the country 'illegally', such as Salim. In order to make a request, citizens and migrants have to go to the Public Centre for Social Assistance (PCSA; referred to as 'the welfare office' throughout this article) of the municipality where they live. Based on a case worker's examination of the situation and on administrative guidelines[1] communicated by the Ministry of Social Integration, a decision is made by the local Welfare Board[2], and the applicant is notified of the decision by post. Should they be unsatisfied with the outcome, people can either ask the Board for a hearing or take their case to court. The court, in turn, can decide to overturn administrative guidelines and ask welfare offices to grant financial assistance to 'illegal migrants'. Because such contestations are almost always brokered by legal aid lawyers or NGOs, they are usually taken straight to court as a strategy for circumventing recent restrictive policies, using human dignity and fundamental rights as justification. While most European countries provide some form of more or less restrictive health-care access to illegalized migrants, Belgium is one of the rare countries that goes so far as to grant financial assistance, although only under very specific conditions (Lafleur & Vintilla 2020).

Based on ethnographic fieldwork within welfare bureaucracies, legal aid offices and NGOs (2018–2019), I ask what kind of claims illegalized migrants make to the state, how they build their cases, and how they understand their (social) rights. I use case studies and interviews with migrants (n=30) to show the crucial role of lawyers, legal clinics and social workers in furthering and shaping migrants' social assistance claims, and sometimes in helping them bring their cases to court. I met the majority of participants through their lawyers or through legal aid NGOs[3]. I introduced myself and my research project before asking if they would permit me to observe the one-on-one meeting with their lawyer or NGO worker; in fact, only a small number asked me to leave the room. Some of them also agreed to meet me at a later date to tell me more about their case. Most of the interviews were conducted in French, some in English: the majority of the research participants had been living in French-speaking cities for several years and were fluent in French. More than half were from French-speaking African countries, some were from Latin America, and a few from other countries. I asked about their experiences with welfare institutions, the rights that they were claiming, and why or how they requested them. Most of them also told me about the different immigration claims that they had made and the various administrative situations that they had experienced over time. Many of the participants had unsuccessfully applied for asylum, and never left afterwards. Some then asked for residency based on the duration of their presence in the country, arguing that they were well integrated into Belgian society[4]. This, however, according to immigration lawyers, is hardly ever granted. Others came to live with a spouse, but lost their resident status after the relationship ended[5]. With the exception of those who recently arrived in the country, most research participants' administrative itineraries read as a succession of more or less successful claims and more or less 'legal' residence statuses. All names were pseudonymized, and the specific details of each case, such as country of origin or medical condition, have been altered to guarantee anonymity. Most of the data were collected in urban areas – rural welfare offices are almost never confronted with illegalized migrants, as the vast majority of them live in large cities.

This article shows that illegalized migrants do not lurk perpetually in the shadows of the state; they also engage with its institutions, ask for public assistance, and deal with the bureaucracies tasked with assessing and delivering the requested services. Much like other residents, they complain about the lengthy, sometimes absurd administrative proceedings, and about their difficult interactions with civil servants, whose role they assumed was to be helping them. Unlike those residents, however, and because most of them have already experienced taking their immigration cases to court, they are regularly in contact with advocacy groups and pro bono lawyers, who play a central role in steering their cases to court. Migrants' stories and their interactions with their lawyers ultimately confirm the legal and political constructedness of 'illegality' (De Genova 2002), but they also highlight how 'legality' can be equally constructed on the basis of welfare entitlements. While dependency is usually associated with a lack of deservingness, leading to the withdrawal of residence permits (Lafleur & Mescoli 2018; Borrelli et al. 2021), in Belgium, severely ill migrants can (sometimes) use dependency as an argument in their favour in immigration courts.

## 2.   Understanding legal and political subjectivities

Gabriella is from Peru. She moved to Belgium with her family more than seven years ago. They entered with tourist visas, and then obtained temporary residence permits based on the fact that Gabriella and her husband both had jobs. Four years later, her husband fell terminally ill and had to stop working. As a consequence, their residence permit was not renewed. They became 'illegal', as Gabriella puts it. With the help of a lawyer, they introduced a new residence permit request, arguing that this was a medical emergency – called a '9ter' procedure (Aliens Act of 1980)[6]. Based on the fact that their immigration case was still pending, Gabriella's lawyer advised her to request financial assistance from her local welfare administration – in addition to medical assistance. When her request was turned down, they took the case to court – and won. Before their immigration claim could be settled, Gabriella's husband died, thereby making the case moot. She started working again and obtained a temporary, one-year resident status on that basis.

Migration scholarship has mainly depicted illegalized migrants as using illegal/informal strategies and either avoiding the state or making claims against it, but in either case staying outside of its institutions (Vasta 2011; Le Courant 2016). Some scholars have nuanced this view by saying that migrants are never fully 'outside' of the state and its laws, using exclusively informal strategies and breaking the law, but that their daily tactics actually include a mixture of illegal/informal strategies and engagement with the state (Scheel 2019; McNevin 2011; Chauvin & Garces-Mascarenas 2020) – as is illustrated by Gabriella's story. Building on their work and on insights from recent studies of statehood and the legal consciousness of ordinary citizens, I ask how illegalized migrants use formal strategies and institutions to get access to social assistance and, sometimes, to 'better papers'.

Anthropological analyses of the state indeed consider the provision of public services and daily interactions between citizens and bureaucrats as the central place where statehood is constituted (Jaffré & Olivier de Sardan 2001; Blundo & Le Meur 2009, Bierschenk & Olivier de Sardan 2014). Focusing on the perspective of ordinary citizens, recent contributions have argued for a 'relational anthropology of the state' (Thelen, Vetters & Benda-Beckmann 2017), one that considers discourses, imaginations, and affective entanglements (Laszczkowski & Reeves 2017). These studies have, however, rarely considered individual migrants' points of view (Eule et al. 2019; Willen 2012a). By shifting the focus from street-level bureaucrats' practices to illegalized migrants' discourses and experiences, this article illuminates the importance of procedural rights – such as the right to due process or to being informed – in order to understand illegalized migrants' expectations of the state.

To be sure, research on legal enforcement does insist on the importance of procedural fairness for people to perceive certain laws or legal actions – such as police work – as legitimate (Sunshine & Tyler 2003; Worden & McLean 2017; Bradford et al. 2014). The ways in which procedural guarantees are perceived and enforced are, therefore, not only key to how people understand laws (Sunshine & Tyler 2003; Nagin and Telep 2017), but they also represent a central aspect of their relationship to the state, determining whether people perceive state actors and their decisions legitimate or not. Illegalized migrants' interactions with welfare administrations show how these procedural guarantees can do more than yield legitimacy; they also shape migrants' expectations of the state and provide them with a way of engaging with its institutions beyond their substantive claims.

Welfare claims also help illuminate a new, and heretofore underexplored, aspect of the political subjectivities of illegalized migrants. I draw on Ewick and Sibley's (1998) three modalities of legal consciousness – 'with', 'without', and 'against' the law – and apply them to migrants' *political subjectivities,* arguing that migrants' expectations of and relationship to the state can be understood in a similar manner. Current scholarship tends to either describe illegalized migrants' daily lives 'without' the state, detailing the informal tactics they engage in to remain hidden from state institutions (Coutin 2005; Psimmenos & Kassimati 2006; Le Courant 2016), or the ways in which they can act 'against' the state by, for example, organizing protests or using fake documents (McNevin 2006; Isin 2008; Vasta 2011). Legal consciousness scholars, however, have recently nuanced such views by exploring the expectations illegalized migrants have when they engage 'with' the law (see, e.g., Schwenken 2013 for a discussion of undocumented migrants' engagement with transnational law). By focusing on expectations of and claims against welfare offices, this article furthers these insights by focusing on illegalized migrants' strategies 'with' the state. They show how illegalized migrants use formal norms and institutions, expect the state to abide by certain rules and provide them with certain services, and sometimes even successfully manage to use these norms and institutions against the state itself. Confirming that illegalized migrants are never fully excluded, but rather always partially incorporated by/in their host countries (Schwenken 2013; Schweitzer 2017; Chauvin & Garcés-Mascarenas 2020), this article contributes to the aforementioned scholarship by exploring the formal strategies migrants employ in order to be included.

These strategies involve using certain state institutions (such as courts) against others (welfare administrations) to remind street-level bureaucrats of the norms that they are expected to enforce. In this game of using 'law against the state' (Eckert et al. 2012), lawyers play a crucial role – just as they do in the making of many other claims (Lakhani 2013; Lejeune & Orianne 2014; Tomkinson 2019). The case of welfare trials, however, helps highlight the complex, embedded relationship between lawyers and the state. Legal aid lawyers are both formally registered with the state and paid by the state through a reward-point system applicable to every case. Not only do they translate migrants'

difficulties into legal terms and actions, but by taking welfare administrations to court, they affect how social assistance policies are enforced. Legal consciousness studies already highlight lawyers' roles in shaping their clients' understandings of law (Lejeune & Orianne 2014), but cases of social assistance also show how lawyers help shape illegalized migrants' political subjectivities, that is, what they expect from the state. These clients are, therefore, not only aware of their rights, but they also expect state bureaucracies to enforce them – which is what their state-paid lawyer will ask the courts for. For all of these reasons, welfare lawyers can be conceptualized as 'brokers of the state' (Dezalay 2018; Bierschenk, Chauveau & Olivier de Sardan 2002; Dezalay 2018). Using the 'Africa' bar of Paris as an example, Dezalay (2018) shows how African lawyers help broker economic deals across continents, between international corporations and African states. In this article, I expand the notion of 'brokers of the state' from the macro scale of international politics to the role of lawyers in shaping everyday understandings of statehood and in implementing public policies at the street level.

Beyond social assistance, these lawyers also use welfare courts to establish their clients' non-deportability – due to health or administrative reasons. In a context where receiving social assistance is often perceived as harming migrants' resident status (Lafleur & Mescoli 2018; Borrelli and al. 2021), I point instead to an exception – one that is still heavily debated both in the courtrooms and in the case law (see Andreetta 2019). At a theoretical level, this aspect reinforces the idea that migrants' agency is mediated and, in part, framed by the state (Chauvin and & Garcés-Mascareñas 2020), but it also takes this view further by showing how illegalized migrants can use state laws and institutions as a strategic resource. Illegalized migrants' agency is, therefore, not only framed by state rules; migrants and their lawyers can, sometimes, use official laws and institutions to their own advantage.

## 3. Asking for social assistance: Belgian laws and institutions

**January 2019, Welfare office**

Salim and I get to the welfare office around 8:30. The entrance is quite small; there are 7 chairs and a reception desk. The clerk asks for Salim's first and last names and for some ID, but he doesn't have any. Other people are given a number, but we're told someone will call his name. We wait for about an hour. I tell Salim about my research, and he tells me about what he had studied in his home country, asks me about my life, where I've traveled, what I do for fun. At 9:20, it is our turn. A social worker takes us to an office. She has a phone and a computer, but she takes notes on a sheet of paper in front of her, where she's already written titles – which, I am guessing, correspond to the different sections of a social report. She asks what we are here for. Salim nods towards me, handing me the doctor's note. I explain that we are here to ask for medical assistance and hand her the document.

The social worker asks who I am and what my relationship to Salim is. I explain that I am a researcher and that I volunteer at the NGO. She then asks how old Salim is, if he is married, if he has kids. I translate questions from French to English, and answers from English to French. She then asks where he sleeps. 'He says in the park, although he sometimes sleeps in the station, especially now', I translate. She asks where he showers, where he eats, charges his phone and washes his clothes. I am aware that she is trying to determine jurisdiction – and hope that she does not end up sending us to another local office based on Salim's answers.

She then enquires when he left Sudan, his home country, and asks a long series of questions about his journey and the countries that he lived in. I am a bit uncomfortable: the questions are precise, intrusive, and she did not give us any explanation regarding what she was going to ask and why. She asks Salim why he left, which means of transportation he used, where he stayed and with whom, how long precisely, how he got to the next country that he visited. Salim keeps his answers to one or two words, regularly saying that he doesn't remember details such as how long exactly he lived in each country. 'Time is not that important to us', he tells me.

> The social worker then asks what kind of degrees or diplomas he has, whether he has been doing undeclared work since he has been here, if he has ever been helped by the welfare administration before. He says he hasn't.
>
> Finally, she asks how his health is. 'I'm not a doctor, she has the paper', Salim states while waving at the desk. 'I ask in order to justify his case to the committee', she answers, in English, to both of us. He explains that the right side of his lower belly hurts. 'All right', she says. She prints out three documents, which she gives him to sign, without explaining anything about them.
>
> Salim immediately signs them without giving me a chance to translate. The first one is a receipt acknowledging that his request for medical assistance was introduced. The second one states that he agrees to provide the administration with every piece of information that they need, including notifying the welfare office of 'any change to his situation', and that he grants them permission to check national and international databases for more information about him. The third one states that the administration has one month to decide on his request, and that if he does not agree with their response, he can go to the Brussels Lower Court of Labour. (Excerpt from fieldnotes, January 2019).

As the 'last safety net' of the Belgian welfare system, social assistance is available to every resident to safeguard human dignity. In contrast to contributory systems such as health or unemployment insurance, social assistance programmes were born out of local charity initiatives – called 'commissions of public assistance' – to alleviate poverty. In 1976, the commissions of public assistance were replaced by Public Centres of Social Assistance (PCSA), regulated by state laws and placed under the funding and supervision of the federal Public Service for Social Integration (PSSI), which is tasked with guiding local institutions in legal and policy implementation.

The aforementioned assistance, however, is not unconditional. In order to qualify for emergency medical assistance, migrants have to fulfil four eligibility criteria, which are defined in a royal decree (AR 1/12/1996)[7]. First, the applicant must be illegally staying on Belgian territory. Second, the applicant must reside in the municipality of the PCSA to which he or she submitted the application. Third, the applicant must be unable to pay for his or her treatment. Fourth, the need for medical care must be urgent and documented by a doctor in a medical certificate template. Financial assistance, on the other hand, can only be granted to those legally staying on Belgian soil – a criterion that can at times be quite ambiguous. Foreign parents of Belgian children, for example, sometimes have to wait months before being granted a residence permit. Should they still be considered illegal residents? What about those whose immigration cases are pending in court? On the aforementioned questions, case law and administrative decisions are not unanimous. Before granting social assistance, the PCSAs are responsible for verifying that the requirements are met, which allows them to apply to the PSSI for the corresponding funding.

Welfare administrations deliver social assistance to both migrants and Belgian citizens in a range of different forms. Citizens can ask for financial assistance, for help paying energy bills, or for specific items such as glasses, diapers or medical treatment. Specific entitlements are mainly contingent upon the migrants' legal status and on the social protections that they already benefit from in their country of origin. Those illegally staying on Belgian soil can benefit from medical assistance. Some residence permits allow those holding them to access financial assistance, while those holding other types of entry permits – such as tourists – are excluded from all kinds of public welfare, including public health care. According to procedural guarantees applicable to all administrative matters, after examining social assistance requests, welfare administrations – or the PCSAs – should deliver written decisions detailing the legal grounds for granting or refusing social assistance, as well as potential avenues for challenging the decision: applicants can either ask for a hearing with the board of representatives within the PCSA or they can petition labour courts, which can overturn or amend decisions from all public welfare administrations.

And in fact, labour courts do sometimes still grant financial assistance to 'illegal' migrants despite welfare law limitations. They justify these interventions on the basis of constitutional guarantees (C. A., n°80/99, 30 June 1999) or international human rights principles such as the prohibition against inhumane or degrading treatment (art 3, ECHR) and the right to an effective remedy (art 13). One category of exceptions is linked to injunctions against deporting people to their home country due to medical, administrative or family reasons, including severe illness, late-stage pregnancy, refusal of the home country to recognize them as citizens, or being the parent of Belgian children. The Constitutional Court indeed found that holding migrants who could not leave the country through no fault of their own to the same standards as those who could would constitute discrimination (C. A., n°80/99, 30 June 1999). More recently, the EU Court of Justice ruled that people who had applied for Belgian residency on the grounds of having a severe medical condition, had had their claim denied by the foreign office, and had brought their case before the immigration courts could also be granted financial assistance. This second category of exceptions is based on their right to effective recourse against administrative decisions, as deporting them to a country where they claim they cannot get proper treatment or leaving them without any resources on Belgian soil would violate their fundamental rights (C-562/13, 18/12/2014). Nevertheless, national courts still have to assess whether the grounds for appeal are substantial and the medical condition is serious enough.

Local welfare offices, however, most often ignore this jurisprudence. In order to benefit from state funding, upon which they heavily rely, they are bound by administrative guidelines issued by the PSSI (Andreetta 2019). These guidelines are mainly based on national laws and are designed to ensure that public funds are allocated according to recent government policies. Without a residence permit, financial assistance can therefore only be obtained by going to court[8].

The above analysis exposes the fragmented nature of the state (Bierschenk 2014) and the fact that specific public institutions may operate according to different and sometimes contradictory logics (Holm Vohnsen 2017; Andreetta 2019). While recent policy changes have been intended to further limit migrants' access to social assistance in an attempt to improve migration control, state courts sometimes implement exceptions in order to protect fundamental rights regardless of – or sometimes even based on – the litigants' 'illegality'.

## 4.   'They always say no': interacting with local welfare offices

**January 2019, welfare office, the end**

Salim shoves the documents angrily into his backpack. The social worker has just explained that he would know in one month whether or not his request is accepted. I ask how he can get medical care in the meantime: 'He can go to the Red Cross', she says, 'but we cannot do anything before the committee decides.' 'Please tell her that I need surgery, tell her that one month is not possible', he orders me. 'If it is urgent, I need a note from the doctor', she responds. 'But I need proof, solid proof, to justify giving out a medical card right now. Then I can go see my superior. But I cannot go empty-handed.' I translate, and Salim calms down a bit, but still mumbles, 'One month, that's not possible', before leaving the room. (Excerpt from fieldnotes, January 2019).

Despite being unhappy with the welfare office, Salim knows that this is, unfortunately, his only hope of getting his medical condition treated. While NGOs can provide basic care and, sometimes, medication in case of emergencies, they are not equipped to perform surgeries nor are they sufficiently funded to provide long-term, regular care. Most of the illegalized migrants that I met therefore relied on NGO doctors for punctual, ordinary healthcare needs, or paid for their treatment themselves. For more serious conditions, however, getting emergency medical assistance from the welfare office is

the only way to access the public healthcare system. This section highlights the obstacles that migrants face and how they perceive ordinary interactions with their case workers, while at the same time illuminating their relationship to the state and its institutions. Like Salim, most of my research participants expected the state to provide and guarantee certain (fundamental) rights, such as the right to medical care, and to provide them with the corresponding services. By introducing a formal request, providing the right documents, and honouring appointments, they were openly engaging with the state and asking it to comply with its own laws – both at a substantive and a procedural level. This shows that their strategies are not only shaped – or limited – by the state and those performing it, but that illegalized migrants are also actively engaging with the state, making social protection claims to its institutions.

Although most of the migrants I met were aware that they could request state-covered medical care, only some of them took advantage of that possibility. Their reticence had a number of causes: the difficult access to welfare offices; their slow, complicated, and multi-step procedures; and the impersonal, often off-putting reception from welfare workers. Far from being restricted to migrants, this phenomenon, referred to as the 'non-take-up of social rights', has been shown to be widespread in other contexts as well (Van Oorschot 1991; Willen 2012b; Warin 2016, 2020), but remains underexplored in Belgium (Dumont, forthcoming). In a 2012 report, Médecins du Monde estimated that 20 per cent of undocumented migrants did not apply for emergency medical assistance (EMA) even when in need of medical treatment (Giladi & Andreetta, forthcoming)[9]. Some give up along the way – around 15 per cent of those introducing a request, according to Giladi (2018) – while another 10–15 per cent see their claims rejected and are unaware of the possibility of appealing or do not dare appeal the welfare office's decision[10]. Frances, a forty-year-old migrant from Cameroon, put it this way:

> They torture people, even just to go to the hospital. If you arrive even five minutes late, it is over, they cannot hear you out. You have to go to the doctor so that he can sign the certificate. Then you go to the welfare office. You cannot put the certificate in the mail. You have to make an appointment. This takes one month. There, you wait, again. When you finally see the social worker, it is one more month to get your medical card. It is just too many problems; a lot of people cannot take it. (Excerpt from interview, 2019).

By 'torture', Frances is referring to the endless, and sometimes absurd administrative steps that she has to go through – all the while being confronted with civil servants' unwelcoming attitude. As illustrated by the above quote, to ask for emergency medical assistance, migrants have to make a series of appointments: with healthcare professionals in order to fill out a medical certificate; with front-desk civil servants at the welfare office; and finally, with their designated social worker once they have successfully passed the initial examination by the front desk workers. 'Sometimes they are so rude that you would think it is their own money that you're asking for', Frances said in the same interview. Similar difficulties are also reported by citizens asking for minimum income benefits or for other forms of social assistance. Citizens, however, do not have to depend on the welfare office for healthcare provision.

When it comes to EMA, most of the beneficiaries meet with their case worker at least twice: first, to complete the social enquiry so that their claim can be filed and examined; then, one month later, to retrieve the precious 'medical card' that will allow them to consult their designated general practitioner. In case of a medical emergency, migrants are advised to go to the hospital – where urgent cases are treated, but no medication is delivered. If it appears that they do not meet the legal criteria, some such emergency claims can be rejected at the welfare office's front desk. This happens, for example, when people cannot prove that they live within the welfare office's circumscribed territory or if they ask for financial assistance – which most welfare offices generally only grant in strict compliance with ministry guidelines. They are, however, expected to fully examine every request and deliver a motivated, written decision to every applicant. In practice, this is not always the case, as Ali explained:

> I was told I didn't qualify for financial assistance, only EMA. They didn't even file the request – they told me it wasn't worth it, that it would be denied for sure. (Excerpt from interview, 2019).

Ali was therefore deprived of the possibility of challenging the de facto negative decision regarding his financial assistance claim; he does not even have proof that his request was introduced and turned down. In other cases, requests are denied after being examined by the welfare office, in which case the applicants are notified of the decision by post. EMA claims are also frequently rejected if the claimant is still in possession of a valid visa: visas are delivered under the assumption that applicants have sufficient resources to support themselves while in Belgium, and when still valid, they are the equivalent of a legal residence permit, which renders the visa holder ineligible for EMA. Another frequent reason for a negative EMA decision is the fact that someone acted as a guarantor when the person was allowed into the country: the guarantor is financially liable for two years after their guest's visa expires. EMA is, therefore, only granted to those whom the state – that is, welfare administrations – recognizes as 'illegal'. Requests for financial assistance, on the other hand, are almost systematically rejected.

If their applications are successful, people are appointed a case worker with whom they have to meet on a regular basis – every three months if they want to have their medical card renewed, provided that they are sick and have a medical certificate to that effect. For those who manage to successfully claim financial assistance, meetings can be organized every six months or every year, provided that their administrative status does not change. Trésor, twenty-two years old and a young mother, explains:

> They [welfare workers] have really become bureaucrats. They should take the person into consideration, but instead, they take papers out of their desk, or they tell you, 'We don't do this here.' […] I know there are rules, but I don't know them. I was never told what they are. Or what I can or cannot do. They don't tell us how it works. They just give you a list of documents to bring. No documents, no money. That's all. (Excerpt from interview, 2019).

The beneficiaries of social assistance – whether they are undocumented migrants, regular residents, or Belgian citizens – explain that meetings with their case workers generally consist of paperwork – delivering it, exchanging it, receiving it. They are rarely informed of the benefits or the services that they can access. For undocumented migrants whose welfare entitlements are limited to health care, the perception is that claiming certain rights and benefits can be done 'only through a lawyer' (Ali, excerpt from interview, 2019).

The aforementioned examples show that illegalized migrants do not limit themselves to informal survival strategies or refrain from engaging with the state, its institutions, and those performing public services. Salim, Ali, Trésor and Frances went to their local welfare offices, made formal requests, and met with the case workers embodying and performing the state on a daily basis. When asking for social assistance, they expected the state to grant them certain rights and protections – provided that they could comply with a series of requirements and could meet certain administrative criteria. In addition to international standards (Schwenken 2013), irregularized migrants also make use of national laws and institutions: in return for fulfilling the abovementioned requirements, they expect welfare administrations to respect procedural guarantees, such as the rights to be informed, to due process, and to an effective remedy.

This ultimately points to a paradox: illegalized migrants experience the absurdity of the state and its bureaucracies all the time, yet they still expect public servants to behave in a fair and rational way. In order to apply for EMA and gain access to public health care, migrants need a medical certificate from a doctor, to whom they can only go (for free) once EMA is granted. It is a classic Catch-22 scenario. This certificate is supposed to state that they are ill and in need of immediate treatment, yet it can take up to a month for the EMA cards to be delivered. Access to health care is granted to those who can prove that they live in Belgium illegally – by providing an address where the welfare administration can establish their residence. Belgian state institutions are, therefore, officially recognizing them as 'illegal' and delivering (welfare) documents on that basis.

Other scholars have already pointed out that illegalized migrants are never fully 'outside' of the state's jurisdiction, but rather that they play a game of partial incorporation (Sheel 2019; Chauvin & Garcés-Mascareñas 2020). Such incorporation also happens through the implementation of proce-

dural safeguards regulating state–migrant interactions. While sociolegal scholars have mainly focused on bureaucratic practices (Spire 2008; Eule et al. 2019; Andreetta 2019; Vetters 2019), welfare requests help illustrate migrants' expectations of the state, demonstrating that they behave as subjects of the state, holding certain rights and entitled to certain services as long as they comply with administrative demands.

## 5. Taking the administration to court: immigration lawyers as brokers

The law, it is so dark inside. (Mathieu, excerpt from interview, 2019)

I did not challenge their decision because I was afraid to make my situation worse. And that they would bring me back to the border, deport me. (Dieudonné, excerpt from interview, 2019)

If they [the welfare administration] are telling me to pay them back, they must be right. It means I do owe them money. (Nadia, excerpt from interview, 2019)

I went to see Mr. David, to see if it was normal or not. Because he knows the law. He is the one who told me that we should go to court. (Gabriella, excerpt from interview, 2019)

In a context where both welfare and immigration laws are incredibly complex and intersect in the form of exceptions built on case law, illegalized migrants often have very imprecise knowledge of their rights. How people understand and use law in order to achieve certain goals is at the heart of legal consciousness studies, but this body of scholarship mostly looks at citizens' understanding of their rights and the legal pathways they use to claim them (Merry 1990; Ewick & Silbey 1998), and not at non-citizens' legal consciousness. For the most part, these studies focus either on courts or on people's everyday experiences of legality, but some highlight the role of intermediaries in building claims and in shaping the legal consciousness of their clients (Lejeune & Orianne 2014). In the case of illegalized migrants, these intermediaries are often paid or subsidized by the state to help their 'clients' contest administrative decisions in court. This section reflects on how court cases are built by specialized lawyers, NGO workers or, sometimes, civil servants, who help shape illegalized migrants' claims to welfare entitlements and, ultimately, their expectations of the state. These intermediaries can, therefore, be conceptualized as 'brokers of the state' (Dezalay 2018; Bierschenk & Olivier de Sardan 2002): they are recognized, funded and accredited by the state, yet they help people build cases against state bureaucracies and, therefore, try to influence administrative practices.

As in most other areas of social life (Engel 2016), within the welfare–migration nexus, people actually taking their claims to court are the exception. As illustrated in the quotes at the beginning of this section, administrative decisions are often assumed to be correct and final – and in line with the law, as Nadia assumed. Even those who feel like they may have been wrongly denied assistance are often reluctant to take their claims further – because of the energy that it would require, the potential repercussions that they may face, or simply because they do not know how to proceed, as the words of Mathieu, Gabriella, and Dieudonné illustrate. Intermediaries therefore play a central role in shaping cases and bringing them to court.

In the cities where this research took place, legal aid could be delivered in two ways: by barristers who agreed to work within the legal aid framework, and by a small number of charities. Most NGOs do provide initial guidance, but very few have the funding necessary to represent their beneficiaries in court. For lawyers, working within the legal aid framework means that they have to file various forms and documents to prove that their clients qualify for legal aid, which generally means that they do not have the resources to pay for a lawyer. Depending on the kind of procedures that the lawyers engage in, they can then claim a certain number of points for each case. Immigration cases are generally worth a lot of points, as are criminal cases; social assistance claims are worth less. Legal aid

lawyers submit an annual global case report to the bar enumerating their completed cases. The cases and the corresponding points are then checked by external examiners from other bar associations. About a year later, they receive payment from the state, the amount of which is determined by how much each 'point' was worth that year. For lawyers, income from legal aid cases is therefore somewhat unpredictable and always delayed, but it is at least certain, which cannot always be said of paying clients, some of whom may never settle their bills. Migrants often get recommendations from friends, NGOs, or welfare workers wishing to help despite administrative guidelines (Andreetta 2019). They can either contact the attorney of their choice directly and ask if they take legal aid clients, or they can go to the bar association and request that an attorney be appointed for them. Free consultations at the bar association are also organized once a week, at which people can ask for initial advice and have a lawyer assigned if necessary. Lawyers generally specialize in one or several areas of law, and select cases accordingly. Most immigration lawyers deal almost exclusively with immigration cases because of the highly precise and technical aspect of the matter. Social assistance lawyers, on the other hand, usually specialize in all matters related to social security. Others concentrate on small personal claims, including family law disputes, social security issues, criminal and immigration cases. Because they are at the intersection of two rather technical areas of law – welfare and immigration – social assistance cases involving illegalized migrants require a lot of work and research, and only a small number of attorneys feel comfortable handling them.

In welfare court, almost every irregular migrant asking for medical or financial assistance was represented by a lawyer. In the cases that I followed, people explained that they talked to their lawyers or to NGO workers about their difficult financial situations, and requesting social assistance was suggested as a possible solution. Sabrina, for example, comes from the Democratic Republic of the Congo and came to Belgium on a medical tourism visa. Her daughter was ill and could not be treated in local hospitals, so Belgian doctors invited her to Belgium to receive the appropriate treatment. Her husband stayed behind. He works as a civil servant in Lubumbashi and sends her a small amount of money every month, but it is not enough for her to be able to afford her own accommodations. Sabrina has been living with her sister since she arrived four years ago, but she no longer feels welcome – their stay was supposed to be temporary, and Sabrina's daughter – and her specific healthcare needs – are the source of increasingly frequent and heated arguments. With the help of her lawyer, Sabrina asked for permanent residency based on her daughter's medical needs. Her claim is still pending in immigration court. She explains how her case against the welfare administration started:

> I told my lawyer about the difficulties that I was having. I was no longer welcome at my sister's, so I asked if there was a way I could get some help finding a house.… She said I could go to the welfare office and ask for financial assistance. She warned me that they might refuse, and told me to come back with the document they would send me if they did. (Sabrina, excerpt from interview, 2019)

As her lawyer predicted, Sabrina's claim for financial assistance was denied, but the two women agreed to take the case to court. As we walked away after the hearing, Sabrina was optimistic: she had answered all of the judge's questions regarding her daughter's medical care and about her husband's income in the DRC. She felt like she was able to argue for herself and her daughter. One month later, she found out that the court had granted her financial assistance on a provisional basis until her immigration claim could be settled.

In other cases, welfare workers themselves help migrants build their cases for court by telling them that they should ask for financial assistance – even if the 'client' was only requesting health care. They then write reports insisting on the precariousness of the migrant's situation, which they know could help them win in court, and advise them to take proof of the negative decision – which they know the welfare office will deliver in order to comply with administrative guidelines – to a lawyer who can help challenge it (Andreetta 2019). Following her social worker's advice, Nadine took the administration to court and was granted social assistance for the future on the grounds that she was the mother of a child authorized to stay on Belgian soil:

> My social worker told me she would argue my case in front of the board. She did it once when I was still pregnant, but they refused. Then, again, when my daughter was born. They refused

again. Then, she told me to go to court and claim my rights. (Nadine, excerpt from interview, 2019).

Nadine had been living in Belgium for more than five years at the time: first as a student when she came to live with her partner, then as an illegalized migrant after they separated and she was pregnant with his child. A year after her daughter was born, she was finally granted legal residency as a member of the child's family.

Looking at workers' experiences of discrimination, Lejeune and Orianne (2014) insist on the central role of intermediaries, such as lawyers and labour unions, in shaping workers' consciousness of their rights. The same is true for illegalized migrants seeking access to state resources and medical assistance. The aforementioned examples show how such intermediaries not only highlight which legal strategies migrants can use and help argue their cases, but they also show them what they can expect from the state and try to impact policy implementation by challenging administrative guidelines in welfare court. They act as brokers between illegalized migrants and state institutions and, in doing so, contribute to the making of statehood by using courts to challenge administrative practices (Dezalay 2018; Bierschenk & Olivier de Sardan 2002). In addition to steering their 'clients' towards lawyers, civil servants from both welfare administrations and healthcare institutions write various kinds of reports that can ultimately help illegalized migrants win their cases. While social enquiries are meant to determine whether applicants meet the legal criteria for financial or medical assistance, they contain a thorough overview of the applicants' situations: their administrative and immigration status, where they live and what their living conditions are like, how they manage to find food and shelter. These reports can, therefore, help prove migrants' lack of resources and need for assistance in court (Andreetta 2019). Medical reports can help highlight the seriousness of a person's condition, the need for care and, if necessary, the need for treatments that are not available in the applicant's country of origin. All of the above are central to welfare courts' decisions to grant financial assistance to those who are severely ill or to members of their families.

## 6.   Becoming 'more legal'

Both Nadine's and Sabrina's cases fall under one of the exceptions according to which welfare courts can grant financial assistance to illegalized migrants: Nadine is undeportable for family reasons, as the mother of a Belgian child. Sabrina's daughter could be too ill to safely return to the DRC. Their cases are built on demonstrating that deporting them would infringe on human rights principles, and that they should get financial aid while they are 'stuck' on Belgian soil. Because welfare institutions are supporting them and because welfare courts determined that they are undeportable, Nadine and Sabrina somehow become 'more legal' through their engagement with state laws, public lawyers and official institutions. Although such cases do not represent the majority of situations, they still provide an interesting counterpoint to the commonly held assumption that welfare dependency leads to deportability (Lafleur & Mescoli 2018; Borrelli et al. 2021).

In order to examine requests such as Sabrina's, welfare judges have to assess the severity of the claimant's illness, as well as the availability of appropriate treatment in their country of origin. They are often presented with administrative decisions from the Aliens' Office that contain the assessment and recommendations of a medical doctor. Such recommendations, often in favour of deporting the applicant, are only based on paper evidence: the administration's general practitioner establishes a course of treatment and suggests potential alternatives to current medication on the basis of the medical documents submitted by the applicant, and establishes the availability of such treatments and medications in the applicant's home country on the basis of official sources. Migrants' lawyers, on the other hand, often submit statements from their clients' medical specialists detailing the specificities of their care, along with NGO reports criticizing the state of public health care in the home country.

As has already been demonstrated by Fassin and d'Halluin (2005) in the case of asylum determination in France, this leap from political subjectivities to biological evidence also creates new economies of deservingness, where only those 'sufficiently' severely ill are granted residency. This is what Ticktin (2011) refers to as the 'illness clause'. In Belgium, 9ter regularization requests – such as the ones introduced by Gabriella or Sabrina – follow the same humanitarian logic as the 'illness clause' in France.

While Ticktin (2011) describes how both migrants and healthcare professionals appropriate the recent 'politics of care' in immigration policies, this article points to another arena where legality can be negotiated (Olivier de Sardan 1995). Although immigration desks and courts decide independently, a positive decision from a welfare court can serve as a substantial piece of evidence in support of one's application for residency: it proves that a national judge has decided that the applicant cannot be deported. Lawyers therefore regularly use welfare dependency as an argument in (certain) immigration cases – as the following vignette illustrates.

**April 2019, legal clinic**

As I do every Friday, I sit though the morning's appointments at the legal clinic. The people who came in today asked to see a welfare law specialist, but the clinic also deals with other areas of law such as immigration, labour, and family law. Preliminary consultations are free, and a lot of the cases, even when litigated, are pro bono.

Our first 'client' of the day, Mr. Safi, explains that he needs medical attention. He went to the welfare office and they gave him a form – he hands over the standard EMA medical certificate form – but he did not understand what he had to do next to get treated. He also explains that he has problems getting by in Belgium, and asks for leads regarding how to earn money.

- Lawyer: You have to go to the welfare office and give them the certificate. You can go to Doctors without Borders, for example, and they will do it for you. Once you have the certificate, you give it to them [the welfare office]. They should answer within one month. If they don't, we'll put pressure on them and go to court. Remember that EMA covers all of the necessary healthcare treatments, not just what is urgent in the strictest sense.

Mr. Safi explains that he has two children, both of whom are allowed to stay in Belgium. They currently live with their mother.

- Lawyer: So, if you have underaged kids, you cannot be deported, so you can ask for financial assistance. But we will have to prove that you still have a relationship with those kids.

- Mr. Safi: No, I won't ask for social assistance.

- Lawyer: Why?

- Mr. Safi: I don't know how…

- Lawyer: You just go to the welfare office, and when you ask for EMA, you also say that you want social assistance. And then, if you get financial assistance for a while, it becomes an argument for your immigration lawyer, because he can say, 'Look, right now, this guy is costing us money. If we grant him residency, he will work and make money.'

(Excerpt from fieldnotes, April 2019)

Focusing on European citizens, Lafleur and Mescoli (2018) show how recent welfare policies create illegal (European) migrants, such as new Italian migrants in Belgium. Recent citizenship policies do indeed allow resident permits to be withdrawn when people have been relying on non-contributory benefits such as social assistance for too long. Migrants are then served with deportation orders, yet never forcefully removed. Pfirter (2019) and Borrelli et al. (2021) show how non-citizens' right to remain in Switzerland is contingent upon their economic participation, and how receiving social assistance might also cause them to lose their residence permits. The examples in this section, on the contrary, show how third country nationals can, under certain, very specific circumstances, become 'more legal' as a result of their reliance on welfare entitlements.

## 7. Conclusion

Using social assistance claims as a starting point, in this article I have described how welfare claims are constructed on the basis of migrants' understanding of their rights, their expectations of the state and public institutions, and the advice of legal intermediaries.

Despite the notable phenomenon of non-take-up of their social rights, illegalized migrants residing in French-speaking Belgium know that they are entitled to public health care if they are unable to pay for necessary medical treatments. They describe the process of claiming these rights as a difficult one, where front-desk bureaucrats are sometimes unfriendly and where administrative requirements can be so absurd and complicated that they sometimes reach Kafkaesque proportions – an experience shared by the majority of welfare beneficiaries, whether citizens or not. Migrants' discourses and complaints about local welfare administrations, however, also illuminate certain expectations the migrants have of the state – the rights that it should grant to those staying in its jurisdiction, as well as the principles that it should abide by, such as due process.

In a context where illegalized migrants are often seen as either hiding from the state or protesting against its immigration policies, and where their strategies are consistently described as either illegal or otherwise outside the law ('without' or 'against' state laws and institutions), this article explores how some of these migrants relate to the state and try to work with its institutions on an everyday basis. As such, it offers a more nuanced reading of two principal ideas in migration studies. First, against this backdrop, welfare claims demonstrate that illegalized migrants use formal, legal strategies in their host countries – in this case, by claiming social assistance in court. Second, these claims illustrate how receiving welfare does not always work against migrants' applications for residency, and can even, in some cases, help advance their claims (Lafleur & Mescoli 2018). This echoes Ticktin's (2011) study on the illness clause in France. By combining ideas from the anthropology of the state and sociolegal studies, this article ultimately demonstrates what we have to gain by considering law and statehood together, yet not necessarily as coherent and unequivocal: in French-speaking Belgium, illegalized migrants do indeed use state laws and legal strategies against the state – or, perhaps more precisely, they use state courts against welfare and immigration desks in their uphill battle to claim social assistance from the state.

## Acknowledgements

I am grateful to all research participants for their trust and input. This article was written up in the framework of *Migration Politics*' editorial fellowship programme – under the warm mentoring of Saskia Bonjour. Special thanks go to Darshan Vigneswaran, Evelyn Ersanilli, Blanca Garcés-Mascareñas, Sébastien Chauvin, Anne McNevin, Rebecca Franco, Patricia Martuscelli and Stephan Sheel for their insightful comments.

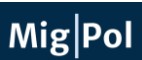

**Funding information.** This research was funded by the Law & Anthropology Department of the Max Planck Institute for Social Anthropology, Halle/Saale, Germany.

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

# Endnotes

[1] Administrative guidelines, also called *circulaires administratives*, provide interpretative guidelines for laws and policy reforms. The amount of state funding that welfare institutions receive depends on the number of requests that they accept while following these guidelines.

[2] Local welfare administrations are governed by a board of representatives, chosen through indirect voting in local elections. These representatives are responsible for making decisions on every single case, based on administrative reports. In practice, in large welfare administrations, the board only discusses a very small number of particularly 'difficult' cases – such as those suspected of fraud or in case of exceptional requests. 'Ordinary' cases are validated automatically, based on the recommendations of the administration, i.e., social workers in charge of the case and their supervisors.

[3] This means that illegalized migrants who are not in touch with any of those organizations were left out of the sample. Those who participated, on the other hand, often assumed that I worked for or with the intermediaries who introduced me to them.

[4] Art 9bis, Aliens Act.

[5] Permanent residency can be obtained after living with a spouse for a period of five years. Until then, migrants are only granted a temporary, one-year residence card, which has to be renewed every year.

[6] The article 9ter of the Aliens Act allows severely ill migrants to apply for a temporary (yet renewable) residence permit in Belgium.

[7] Royal decrees are a source of law; they are written by the government for the purpose of clarifying and helping enforce laws.

[8] On very rare exceptions, welfare offices will grant financial assistance out of their own funds.

[9] A 2016 report states that due to lack of quantitative data, the non-take-up of social rights among all residents of Brussels is impossible to quantify. The same report highlights, however, that across most EU

countries, 'more than one-third of people who are entitled to certain benefits do not receive at least one of them' (Observatoire de la Santé et du Social de Bruxelles 2016: 12, translated by the author).

[10] These numbers are based on the researchers' observations (Giladi & Andreetta, forthcoming). Official statistics are based on the amount of federal funding allocated to local welfare administrations; therefore, only 'granted requests' are counted.