# Peer review of "Engaging with the State. Illegalized migrants, welfare institutions and the law in French-Speaking Belgium"

_Migration Politics_

## Round 1 · Referee Report · Sébastien Chauvin (Referee 1) · 2021-11-28

Strengths

  1. The paper provides an original take on how illegalized migrants make use of existing laws and formal venues as strategic resources to claim health benefits, expecting procedural fairness (while meeting many obstacles)
  2. The paper qualifies the assumption that welfare "dependency" is always a source of legal undeservingness.
  3. The paper explores the legal consciousness of (unauthorized) non-citizens or "ordinary subcitizens", while most studies of legal consciousness focus on citizens.

Weaknesses

  1. Although instances of working "with the state" are well emphasized, this risks giving the impression that article's informants can be reduced to this strategy. Perhaps a few hints here and there of how the "with the state" strategy might be combined with others in some instances may counterbalance this impression (although the article already does give examples of the non-take up of social rights).

  2. The last section (n.6) on the possibility that being a benefits claimant can enhance their legal deservingness remains rather underdeveloped, so that proofs brought in to support it could be perceived as still anecdotal or mostly virtual. The section could be expanded a bit (so as not to read as a side point or an afterthought) as the argument is original.

Report

This is a fine draft making original arguments about the relationship between illegalization, health-related deservingness, and chances of eventual legalization.
The article mostly engages with the anthropological literature on the state, the socio-legal literature on legal consciousness, and the migration studies literature on legal deservingness.
Perhaps the article does not engage enough yet with the existing literatures on health-related deservingness, especially that related to legally precarious migrants. I'm thinking in particular of the work of Sarah Willen.
Also, only passing mention is made of Myriam Ticktin's study of the illness clause in France and there is no mention of Didier Fassin and Estelle D'Halluin's work, especially this article:
https://anthrosource.onlinelibrary.wiley.com/doi/abs/10.1525/aa.2005.107.4.597?casa_token=TvltTehgavoAAAAA:iQot95S7tovSi9JXxlqXLc5AY7bCdjQ-Dxd5pa0FLbUmDN5VE-YnUcHLdA78qMimHjgdErxXdqptyQEF
Although these pieces deal mostly with asylum seekers, their object is close enough to that of the present article (including the role of intermediaries and guarantors of vulnerability) so as to warrant more engagement with them earlier in the literature review (rather than just occasionally in section 6).
Finally, the legal consciousness literature does include some works on the legal consciousness of undocumented migrants (rather than that of "ordinary citizens"), which the author could do a better job reviewing.
For an example, I'm thinking of the following piece by Helen Schwenken:
https://onlinelibrary.wiley.com/doi/10.1111/imig.12118
Other occasional remarks:
p.2, in the first sentence of the first paragraph, not sure placing the sentence part "Mainly focused ... legal status" ahead of the main proposition is colloquial in English.
p.2, in the last but one paragraph, the term "administrative guidelines" is ambiguous: does this refer to formal guidelines or to an administrative decision in a given case?
p.7, 1st paragraph, not clear if "The first type of exceptions" refers to previously mentioned "constitutional guarantees" or starts a new list not related to the two "bases" mentioned above.
p.13, the word "emergent" should be replaced with "urgent", I believe.
p.14 in the last paragraph of the conclusion, the author mentions "a more nuanced reading of two ideas" but it was not entirely clear to me what these two ideas were.
Finally, in the same paragraph, I wasn't entirely convinced by the division between "laws" and the "state", where the authors argues that laws can be deployed against the state. Although this is true, these laws are also part of the state, so that it could equally be said that the state is strategically used against the state.

Requested changes

see in report & weaknesses section

  • validity: high
  • significance: high
  • originality: top
  • clarity: high
  • formatting: perfect
  • grammar: excellent

Author:  Sophie Andreetta  on 2022-01-31  [id 2133]

(in reply to Report 1 by Sébastien Chauvin on 2021-11-28)

Dear editor, Dear Sebastien (if I may),

Many thanks for your very helpful comments and feedback. Here is how I incorporated your suggestions. I hope that you will find the revised version of the article acceptable.

  1. Although instances of working "with the state" are well emphasized, this risks giving the impression that article's informants can be reduced to this strategy: I nuanced one of the sentences in the introduction, and added a few more hints towards the fact that migrants use a mix of “with, without and against” the state strategies throughout the paper (on page 7 and on page 12).
  2. The last section (n.6) could be expanded a bit (so as not to read as a side point or an afterthought) as the argument is original. I’ve used some of the literature suggestions (particularly Ticktin and Fassin’s texts) in order to thicken the argument in this section. I have also referred back to some of the case studies mention elsewhere in the text.
  3. Perhaps the article does not engage enough yet with the existing literatures on health-related deservingness, especially that related to legally precarious migrants. I’ve added a few sentences in the literature review section (page 4) – mainly pointing out what seemed relevant to my overall argument about migrants’ relationship to the state. I have also used some of the references throughout (on welfare deservingness especially) including in section 6.
  4. Smaller comments • 4. p.2, in the first sentence of the first paragraph, not sure placing the sentence part "Mainly focused ... legal status" ahead of the main proposition is colloquial in English : The sentence has been modified. • p.2, in the last but one paragraph, the term "administrative guidelines" is ambiguous: does this refer to formal guidelines or to an administrative decision in a given case? I have included a footnote on administrative guidelines, and some precisions in the text. • p.7, 1st paragraph, not clear if "The first type of exceptions" refers to previously mentioned "constitutional guarantees" or starts a new list not related to the two "bases" mentioned above. The paragraph has been clarified. • p.13, the word "emergent" should be replaced with "urgent", I believe. Done. • p.14 in the last paragraph of the conclusion, the author mentions "a more nuanced reading of two ideas" but it was not entirely clear to me what these two ideas were. The sentence has been clarified. • Finally, in the same paragraph, I wasn't entirely convinced by the division between "laws" and the "state", where the authors argue that laws can be deployed against the state. The second part of the sentence clarifies this: “using state courts against immigration desks”.

---

## Round 2 · Author Response

Dear editor,
Dear Sebastien (if I may),

Many thanks for your very helpful comments and feedback.
I hope that you’ll find the revised version of the article acceptable.
You'll find a list of the changes I made below.

Kind regards,
Sophie

---

## Round 2 · List of Changes

1. Although instances of working "with the state" are well emphasized, this risks giving the impression that article's informants can be reduced to this strategy: I nuanced one of the sentences in the introduction, and added a few more hints towards the fact that migrants use a mix of “with, without and against” the state strategies throughout the paper (on page 7 and on page 12).
  2. The last section (n.6) could be expanded a bit (so as not to read as a side point or an afterthought) as the argument is original. I’ve used some of the literature suggestions (particularly Ticktin and Fassin’s texts) in order to thicken the argument in this section. I have also referred back to some of the case studies mention elsewhere in the text.
  3. Perhaps the article does not engage enough yet with the existing literatures on health-related deservingness, especially that related to legally precarious migrants. I’ve added a few sentences in the literature review section (page 4) – mainly pointing out what seemed relevant to my overall argument about migrants’ relationship to the state. I have also used some of the references throughout (on welfare deservingness especially) including in section 6.
  4. Smaller comments • 4. p.2, in the first sentence of the first paragraph, not sure placing the sentence part "Mainly focused ... legal status" ahead of the main proposition is colloquial in English : The sentence has been modified. • p.2, in the last but one paragraph, the term "administrative guidelines" is ambiguous: does this refer to formal guidelines or to an administrative decision in a given case? I have included a footnote on administrative guidelines, and some precisions in the text. • p.7, 1st paragraph, not clear if "The first type of exceptions" refers to previously mentioned "constitutional guarantees" or starts a new list not related to the two "bases" mentioned above. The paragraph has been clarified. • p.13, the word "emergent" should be replaced with "urgent", I believe. Done. • p.14 in the last paragraph of the conclusion, the author mentions "a more nuanced reading of two ideas" but it was not entirely clear to me what these two ideas were. The sentence has been clarified. • Finally, in the same paragraph, I wasn't entirely convinced by the division between "laws" and the "state", where the authors argue that laws can be deployed against the state. The second part of the sentence clarifies this: “using state courts against immigration desks”.

---

## Editorial Decision

unknown